**Data Availability Statement:** All relevant data are within the manuscript.

**Funding:** DMC IOS-1655113. National Science Foundation (USA) NSF.gov The funders had no role in study design, data collection and analysis,

# The effect of odor enrichment on olfactory acuity: Olfactometric testing in mice using two mirror-molecular pairs

**Alyson Blount**[ID]**, David M. Coppola**[ID]*

Department of Biology, Randolph-Macon College, Ashland, Virginia, United States of America

* dcoppola@rmc.edu

## Abstract

Intelligent systems in nature like the mammalian nervous system benefit from adaptable inputs that can tailor response profiles to their environment that varies in time and space. Study of such plasticity, in all its manifestations, forms a pillar of classical and modern neuroscience. This study is concerned with a novel form of plasticity in the olfactory system referred to as induction. In this process, subjects unable to smell a particular odor, or unable to differentiate similar odors, gain these abilities through mere exposure to the odor(s) over time without the need for attention or feedback (reward or punishment). However, few studies of induction have rigorously documented changes in olfactory threshold for the odor(s) used for "enrichment." We trained 36 CD-1 mice in an operant-olfactometer (go/no go task) to discriminate a mixture of stereoisomers from a lone stereoisomer using two enantiomeric pairs: limonene and carvone. We also measured each subject's ability to detect one of the stereoisomers of each odor. In order to assess the effect of odor enrichment on enantiomer discrimination and detection, mice were exposed to both stereoisomers of limonene or carvone for 2 to 12 weeks. Enrichment was effected by adulterating a subject's food (passive enrichment) with one pair of enantiomers or by exposing a subject to the enantiomers in daily operant discrimination testing (active enrichment). We found that neither form of enrichment altered discrimination nor detection. And this result pertained using either within-subject or between-subject experimental designs. Unexpectedly, our threshold measurements were among the lowest ever recorded for any species, which we attributed to the relatively greater amount of practice (task replication) we allowed our mice compared to other reports. Interestingly, discrimination thresholds were no greater (limonene) or only modestly greater (carvone) from detection thresholds suggesting chiral-specific olfactory receptors determine thresholds for these compounds. The super-sensitivity of mice, shown in this study, to the limonene and carvone enantiomers, compared to the much lesser acuity of humans for these compounds, reported elsewhere, may resolve the mystery of why the former group with four-fold more olfactory receptors have tended, in previous studies, to have similar thresholds to the latter group. Finally, our results are consistent with the conclusion that supervised-perceptual learning i.e. that involving repeated feedback for correct and incorrect decisions, rather than induction, is the form of plasticity that allows animals to fully realize the capabilities of their olfactory system.

decision to publish, or preparation of the
manuscript.

**Competing interests:** The authors have declared
that no competing interests exist.

## Introduction

The task of natural intelligence, like artificial intelligence, is to correctly interpret environmental information in pursuit of certain goals. This requires sensory systems that can capture mission-critical data in a world teeming with stimuli that are irrelevant, ambiguous or underspecified. Universal and enduring sensory statistics, the knowledge of which can promote survival, are often "hardwired" into neural systems. For example, natural scene statistics which have a preponderance of cardinal contours are matched by a mammalian visual cortex with an innate preponderance of neurons responsive to vertical and horizontal edges [1–3].

However, it is impractical to program all needed sensory information into intelligent systems, pointing to the survival advantage of learning or, more broadly, neural plasticity. One of the most studied forms of plasticity, that involving "critical periods" during development, enjoys the advantages of both adaptability to a unstable environment and resistance to alteration after formation. Mammalian vision, again, provides a classic example in ocular dominance columns of visual cortex, which are only mutable during a prescribed period of early life [4, 5]. Adult plasticity, including the various forms of memory, round out the modes of information processing and storage with these later forms representing the most evanescent.

Here we are concerned with "induction" a mode of information processing in the olfactory system that has been likened to perceptual learning, a form of implicit memory by which abilities in a sensory task are improved upon with or without the necessity of feedback [6–8]. The classic form of induction involves anosmias. In these deficits, which are widespread in humans, there is an inability to smell a particular odor in a subject that otherwise has a normal sense of smell [9]. However, in certain cases non-smellers can be transformed to smellers by repeated exposure to the odor for which they were initially anosmic [10]. In animals, a related phenomenon has been reported in which subjects that initially are unable to discriminate like-pairs of odors develop the ability to do so without feedback after several days of enrichment [11]. These unsupervised forms of olfactory learning, that have been associated with peripheral or, at least, low-level plastic changes in the nervous system, will be referred to as induction in keeping with Wysocki and colleagues [10]. The modes of odor exposure that have typically precipitated induction will be referred to as "enrichment." Usually, these regimens have involved one or a small number of purified odorants delivered to subjects at relatively high concentrations with exposures lasting from several days to several months [10, 11].

Relatively few studies have rigorously compared the effects of passive (unsupervised) enrichment and active (supervised) enrichment on behavioral acuity for the enriched odor. Here we combine operant olfactometric testing with extensive replication to ask if enrichment with the enantiomers (mirror-molecules) of limonene and carvone cause improved behavioral acuity through either heightened ability to discriminate the stereoisomers of these odors or to detect one stereoisomer. Though we found no evidence of induction—no change in discrimination or detection thresholds—after either active or passive enrichment, we did obtain among of the lowest thresholds ever measured for any species. The implications of these data for interpreting thresholds in other studies including the comparison of mice and humans and for assessing the role of perceptual learning in olfactory information processing are discussed.

## Methods

### Animals

All animal procedures were approved and supervised by the Randolph-Macon College Institutional Animal Care and Use Committee and comply with the "Guide for the Care and Use of Laboratory Animals" (8th Edition, National Academies Press USA). Mice were kept in an

approved animal room maintained on a 12hr/12hr light cycle with mouse chow available *ad lib*. Subjects were females of the CD-1 outbred strain obtained from Charles River Laboratories (Wilmington MA, USA) at 56–60 days of age and used in the study up to 7 months of age. This strain was chosen because it has been the most common target of olfactometric studies, in general, and thresholds have been obtained previously from the strain using similar stimuli and procedures [12].

Thirty-six mice completed threshold testing. Several other subjects were trained to some degree and later removed from the study because of their inability to perform the operant task consistently. Beginning 5–7 days before the beginning of operant training, mice were placed on a 1 ml daily ration of water bringing their body weight to below 85% of their free-drinking weight. Thereafter, daily water rations were adjusted individually to between 1–1.5 ml to maintain each mouse as close to its 85% *ad lib* weight target as possible.

## Stimuli

The stimuli were the enantiomers of limonene and carvone which were obtained from Sigma-Aldrich (St. Louis, Mo, USA) at the highest available purity (all > 96%). These odorants were chosen for several reasons: First, enantiomers have identical physical properties, other than chirality, assuring that discrimination cannot be based on concentration or other properties unrelated to odor quality. Second, as noted above, psychophysical measurements of these compounds have already been reported for CD-1 mice [12] based on data obtained from nearly identical olfactometric equipment (Knosys, Tampa, FL, USA). Third, though limonene and carvone are structurally similar, previous studies have shown that discrimination of limonene enantiomers but not carvone enantiomers can be induced in rodents through passive enrichment [13]. Fourth, we assumed that one of the tasks we chose, discrimination of a mixture of enantiomeric isomers from a pure isomer, would be very difficult, thus leading to rapid threshold determination requiring a minimum of odor exposure.

Stimuli were produced by sampling the head-space above odor/mineral-oil (CVS brand) mixtures contained in the odor reservoirs of the Knosys system. Owing to the difficulty of accurately estimating the gas-phase concentration of odor/mineral oil mixtures without direct chromatographic measurement and the fact that relative concentrations were sufficient to test for odor induction, concentrations, except those in Table 1 and Fig 7, are given in the liquid-phase units of vol/vol ppm [14].

The required concentrations were made by serial dilution of ml volumes starting with 1,000 ppm (0.1 vol/vol%) stocks solutions. A key advantage of serial dilution is that the

**Table 1. Median liquid-phase odor concentration (Conc.) thresholds, their 95% Confidence Limit (CL), and vapor-phase odor concentration estimates.** Control and odor exposed data have been pooled. All values are in ppm.

| | | | | Limonene | | | | |
|---|---|---|---|---|---|---|---|---|
| | **Discrimination** | | | | | **Detection** | | |
| Liquid Conc. | Lower CL | Upper CL | Vapor[†] Conc. | | Liquid Conc. | Lower CL | Upper CL | Vapor[†] Conc. |
| 5.5E-11 | 1 E-12 | 1 E-10 | 6.16E-15 | | 5.5E-13 | 1 E-16 | 1 E-11 | 6.16E-17 |
| | | | | Carvone | | | | |
| | **Discrimination** | | | | | **Detection** | | |
| Liquid Conc. | Lower CL | Upper CL | Vapor[§] Conc. | | Liquid Conc. | Lower CL | Upper CL | Vapor[§] Conc |
| 1 E 0 | 1 E-5 | 10E 0 | 7.94E-6 | | 5.5E-4 | 1 E-5 | 1 E-2 | 4.37E-9 |

† Values of Henry's law parameters from Cometto-Muniz et al., 2003.

§ To obtain carvone values, limonene values were adjusted for difference in VP between these two odors.

experimenter is measuring milliliter quantities, except for the starting stock solution, not microliter quantities of odor stimuli and solvent. Importantly, any errors in measurement are merely additive across dilution steps! Thus, simple mathematical simulations that we performed to introduce random or systematic errors into each of 19 serial dilutions measurements produced acceptably low final disparities ($< 0.35$ log units) between nominal and simulated concentrations (data not shown).

Fresh odorants were purchased every two months and new serial dilutions were made up every two weeks (at the longest) during active testing which took place over approximately 18 months.

## Testing

Olfactory testing employed a commercial air-dilution olfactometer (Knosys, Inc. Tampa, FL, USA). This computer-automated system shapes the behavior of subjects to either insert or withdraw their snout from a sampling port to show differentiation of a water-rewarded stimulus (S+) from a non-water-rewarded stimulus (S-). This apparatus has been widely used and validated for odor discrimination and detection testing (see Discussion). Details of its operation with mice have been thoroughly described elsewhere [24]. Briefly, subjects were trained, using standard operant procedures, to initiate a trial by placing their snout in the odor port, and thus breaking a photobeam. This triggered the delivery of a stimulus, either an odor or clean air (blank) for two seconds. After a minimum sampling time of 0.5 seconds, the subject was required to respond by either licking a tube inside the port to obtain a water reward, an S + trial, or withdrawing their snout from the port to register the expectation of a no reward, as an S- trial. Both licking on an S+ trial or not licking on a S- trial were scored as correct responses. Failing to lick on a S+ trial or licking on a S- trial were scored as incorrect responses. The order of S+ and S- delivery were shuffled for each 20 trial block with the prohibition of three of the same stimuli in a row with further constraints of equal numbers of S + and S- trials per blocks. Mice were run in two or occasionally three daily sessions of up to five blocks (100 trials) with the exception that seven blocks were allowed after failure to reach criterion in two five-block sessions (see below).

## Thresholds

The 36 mice that completed the study went through at least one mixture discrimination threshold and one detection threshold measurement. Half of the subjects were tested on limonene and the other half were tested on carvone. Eleven mice from the limonene group were part of a "pre/post" design and were tested for enantiomer mixture discrimination before and after an enrichment period. The other seven mice in the limonene group were either not enriched controls (n = 2) or were tested only after an enrichment period (n = 5). For the carvone group, the pre/post design was dispensed with: nine subjects serving as controls and nine receiving odor enrichment prior to a single mixture discrimination and detection measurement.

Single-isomer/mixed isomer discrimination was of the form: **S** vs. **R** + **S**, where **R** represents 1,000 ppm stock of the right-handed isomer and **S** represents 1,000 ppm stock of the left-handed isomer of limonene and carvone. The discrimination threshold was defined as the lowest concentration of **R** stock diluted in **S** stock that could be distinguished from pure **S** stock. For example, a 100 ppm discrimination threshold would mean that the mouse could discriminate a 1/10 dilution of **R** stock in **S** stock from the pure **S** stock. We chose this task, in which the **R** isomer is the target and the **S** isomer the background, because we thought it would be exceptionally difficult given our prior experience testing mixture discriminations in mice with

other structurally less similar odorants [15]. Detection threshold was defined as the lowest concentration of **R** isomer that could be discriminated from a mineral oil blank.

A modified descending method of limits was used for all threshold measurements. Threshold testing began by training a subject to discriminate pure **R** stock from pure **S** stock. Following this, **R** + **S** concentrations were decreased in 10-fold increments. For efficiency, this was changed to 100-fold increments after a mouse passed criterion on three successive concentrations. When a subject failed on a particular concentration, a 10-fold higher dilution was tested to provide a more precise (within a factor of 10) measurement of threshold.

In order to pass at a particular dilution, a subject had to achieve a minimum of 85% correct in the last two blocks of a session or to have a minimum average of 85% correct for the last 3 blocks. If a subject failed to reach criterion in the first five-block session, she was given a second five-block session. If the subject failed again, she was given a third and final-seven block session to meet the 85% criterion. These threshold criteria were the product of extensive preliminary testing in which it was established that the common practice of basing threshold measurements on the outcome of only a small number of trials consistently underestimates olfactory acuity (see discussion).

## Control procedures

A regular program of olfactometer cleaning was implemented throughout these studies to minimizes odor contamination. All tubing was replaced at regular intervals and whenever different odors were tested. Glass parts that came in contact with odor stimuli, were regularly cleaned with 70% alcohol and dried for at least 24 hr in an a 60°C oven reserved exclusively for this purpose. Operators wore latex gloves whenever handing any part of the olfactometer.

Special control procedures were implemented starting at dilutions below $1 \times 10^{-4}$ ppm, to make sure that each subject was making discriminations based on the test odors and not extraneous odors or other sensory cue. For each session in which the subject passed the criterion they also had to pass a control test before they could move on to higher dilutions. This test consisted of two blocks (40 trials) at the end of a five-block session in which either pure mineral oil was placed in the S+ and S- channels (detection tests) or the stimulus not associated with the water-reward for that session was placed in both the S+ channel and the S- channel (discrimination tests). In either case, these measures would have made discrimination of the S + and S- channels impossible on the basis of the experimental stimuli. If performance on both blocks of these control trials fell below 60% correct responding, then it was concluded that the subject's prior discrimination of S+ and S- channels was based on differentiating the experimental stimuli rather than uncontrolled stimuli such as valve sounds, somatosensory cues, or extraneous odor contamination. In the rare instances in which a mouse failed the control test, some combination of playing a radio loudly during testing, scrambling the S+ and S- odor channels among the eight channels of the olfactometer, or replacing all the tubing and remixing stimuli brought the subject's behavior back under stimulus control [14].

## Enrichment

Three types of environmental enrichment were used in these studies. In the first regimen, termed "active," four mice performed in the olfactometer five days a week for three weeks. On each of the five enrichment days the subjects were tested on 100 trials (five blocks of 20 trials each) in which they were tasked with discriminating the stock solution of **R**-limonene (1,000 ppm) from the stock solution of **S**-limonene (1,000 ppm). As part of the pre/post design, these four mice had limonene mixture discrimination threshold measured before and after enrichment and **R**-limonene detection thresholds measured after enrichment as described above.

In the second regimen, termed "passive," mice were enriched with either both limonene enantiomers (four mice; duration three weeks; in pre/post design) or both carvone enantiomers (nine mice; duration three weeks to three months; between-subjects design) by mixing odor in their food which was provided *ad lib*. Odorized food was made by first grinding mouse chow in a standard food processor. A 0.1% v/v solution of each enantiomer diluted in distilled water was then mixed in a one to two ratio (v/v) with ground chow. This slurry was then formed into blocks by filling plastic ice trays which were allowed to dry overnight in a fume hood. The resulting food blocks were stored at 0 deg C° and thawed as needed to replace the food supply of the enriched mice which were housed in a fume hood during the enrichment period. Scented food was replaced with freshly-thawed, scented food every three days.

A third group (three mice), part of the pre/post design, was exposed to an 'exercise' enrichment regimen in an effort to control for the difference in physical and cognitive demands between the active odor enrichment and passive odor enrichment group. These mice were not odor enriched but rather were transferred for 1 hr per day for two weeks into a standard rat cage (18.5 x 10 x 8 cm) that contained a running wheel and an assortment of randomly scattered small objects which were relocated in the cages daily. As for the other pre/post groups, limonene mixture discrimination thresholds were measured before and after enrichment and **R**-limonene detection thresholds were measured after enrichment as described above.

## Statistics

Non-parametric statistics were used for two-tailed hypothesis tests with the alpha level set at $p \leq 0.05$. The Mann-Whitney's U tests was used for independent samples and the Wilcoxon's test was used for matched pairs (Prism 8.2.4, GraphPad). Wherever possible, data from different subgroups were pooled in order to increase statistical power if they reasonably could be assumed to have been samples drawn from the same underlying population. In the majority of cases, pooling involved subgroup that were not significantly different and thus were assumed to be independent samples from the same underlying population. The figures which show the results of hypothesis tests also show medians and their 95% confidence interval (CIs; Prism 8.2.4).

## Results

Naïve mice rapidly learn to discriminate 1,000 ppm concentrations of the **R** and **S** enantiomers of limonene and carvone. Fig 1A shows the % correct responses in the first several blocks of testing for two naïve mice in the limonene group and two naïve mice in the carvone group. Collectively, for the 9 naïve mice (not previously odor exposed) in the limonene group, the median number of blocks to meet criterion was 17. Coincidentally, this was the same value for the nine naïve mice in the carvone group. However, as can be seen in Fig 1A, some mice were responding well above chance levels within a few blocks of starting the enantiomer discrimination task (e.g. Fig 1A, Mouse 28).

Contrary to our expectation that **S** vs. **R** + **S** enantiomer discriminations would be difficult, the majority of our naïve and odor enriched mice displayed remarkable capabilities at this task. Fig 1B shows all the block scores leading to naïve Mouse 31's final mixture threshold determination at $1 \times 10^{-12}$ ppm. These data illustrate several characteristics which were common to nearly all the mice in the study. First, the initial discrimination of **S**-stock from 1,000 ppm **R**-stock diluted in **S**-stock often took two or three sessions of five to seven blocks each before mice mastered the task. Decreasing the concentration gave the appearance of a qualitative change in stimulus in that responding typically fell back to chance levels for a block or more. Third, as concentration approached threshold, % correct responding became quite

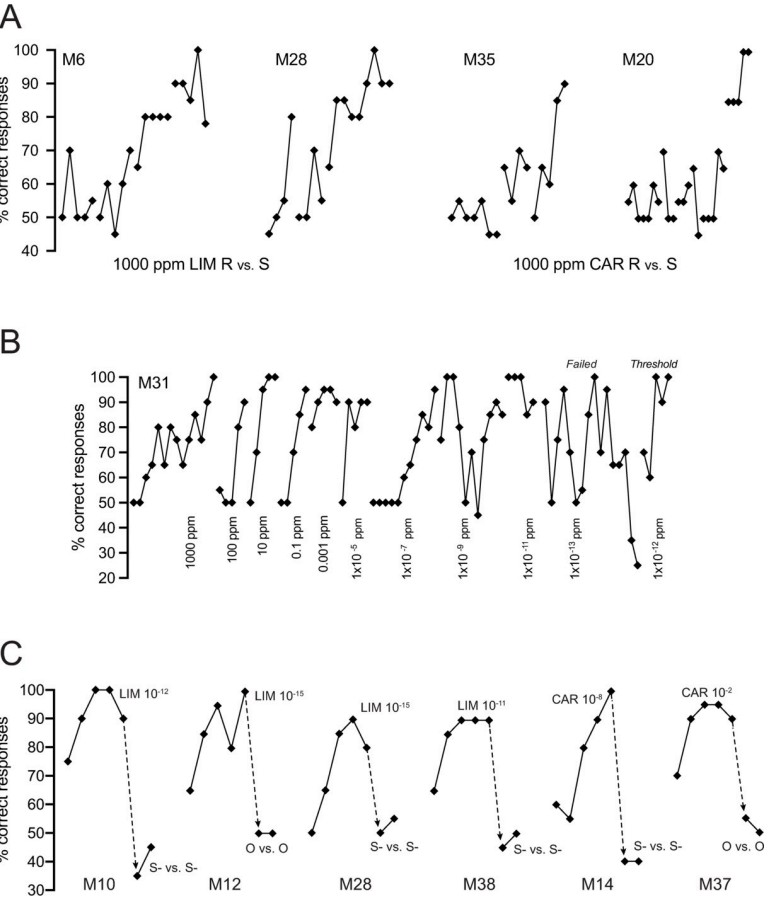

**Fig 1. Percent correct responses (diamond symbols) during consecutive blocks of 20 trials.** The plot should be read from left (first block) to right (last block). Lines between symbols connect blocks in the same session. **A.** Initial blocks for four mice, two discriminating 1,000 ppm limonene (LIM) **R** vs. **S**, two discriminating 1,000 ppm carvone (CAR) **R** vs. **S**. Note that all mice except M20 made progress toward discrimination within a session or two. **B.** Entire record for Mouse 31 on her path to limonene discrimination threshold. Discrimination took the form **S** vs. **R + S** where **R** and **S** are the two stereoisomers of limonene at 1,000 ppm stock concentration in mineral oil. The stated concentrations are consecutive dilutions of **R** stock in **S** stock (left to right). Note that % correct responses often fell to chance levels when concentration was decreased. As was typical, % correct responses became quite unstable near threshold (see failure at 1 x $10^{-13}$ ppm). Note that for first five blocks of concentration 1 x $10^{-7}$ the mouse was indiscriminately responding on every trial. **C.** Examples, in six mice, of control procedure to assure subjects were discriminating on the basis of experimental stimuli. In the case of S- vs. S- the mouse was given identical S- stimulus on every trial. In the case of O vs O, mouse was given identical odorless mineral oil on every trial. Note that highly successful discriminations dropped to near chance levels of correct responding (50%) when the identical stimulus was associated with rewarded and unrewarded trials (dashed arrows).

unstable, often alternating from high success rates to chance success rates, block to block. This instability in responding often caused the mice to fail the threshold criterion despite the fact that they showed near perfect responding for a block or two.

A perennial concern common to olfactometers such as the Knosys system is that valve sounds (or vibrations) or other factors like odor contamination in the odor channels can become the unintended basis of discriminating the rewarded from the non-rewarded channel. As a control for this possibility all mice discriminating at high levels (low concentrations) were tested by presenting the non-rewarded stimulus or neat mineral oil in both the S+ and S- channels (see methods). Fig 1C shows the results for six mice whose high % correct responding fell to chance levels when confronted with one of these control pairs.

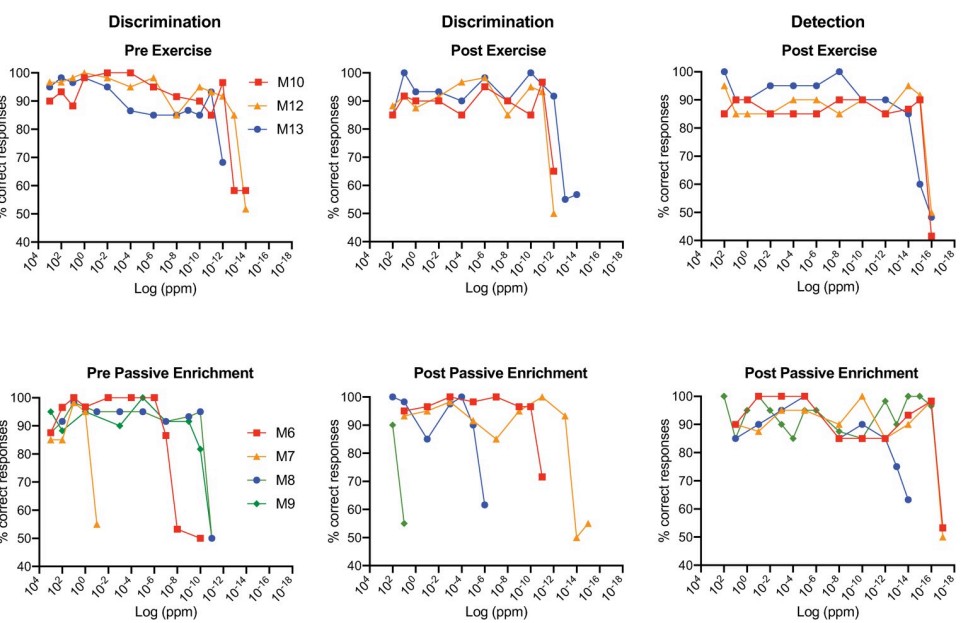

**Fig 2. Performance of mice discriminating (left-most two columns) or detecting (right column) various dilutions of limonene.** Discrimination was as noted in Fig 1. Detection took the form **R** vs. mineral oil. The top row displays the results of three control mice which were not enriched with odor but had daily "exercise" enrichment (see details in Methods and Materials). The bottom row displays the results of four mice before (pre) and after (post) three weeks of enrichment by mixing both stereoisomer of limonene in their food. Each symbol represents the % correct responses for an individual mouse in the last two or three blocks of trials (see details of threshold criteria in Methods and Materials).

## Limonene pre-post design

The initial (pre-enrichment) threshold measurements for what were to be the active enrichment group had to be excluded because less stringent threshold criteria were in force causing what we later learned to be an underestimation of olfactory abilities (see discussion). Thus, only the post-enrichment thresholds are reported for these four mice (M1-4) using the revised criteria.

Fig 2 shows criterion responding until threshold for the three mice in the exercise group and the four mice in the passive enrichment group. As noted above, some mice in every group could discriminate enantiomeric mixtures or detect the **R** stereoisomer of limonene at very low concentrations (down to 1 x $10^{-16}$ ppm in a few cases). However, overall thresholds for the enrichment group were highly variable (> 16 log unit range) which obscured group differences. Particularly challenging was the fact that little test-retest reliability was in evidence: compare M9's thresholds in the passive exposure group before and after enrichment, for example, which differed by more than 10-log units. The exercise group data were far less variable but there was no obvious effect of exercise on the median thresholds (Fig 2).

Fig 3A illustrates the individual thresholds for mice in the pre/post design with arrows depicting the initial and final threshold. There was little change in mixture discrimination thresholds with exercise (two increasing and one decreasing), however, all three mice had lower thresholds in the detection than the discrimination task suggesting, as expected, that the former task was easier than the latter. For the passive enrichment regimen, the four subjects displayed large and inconsistent pre-to-post changes in threshold with two mice dramatically increasing mixture discrimination thresholds (5-log units and 11-log units respectively) and two mice dramatically decreasing mixture discrimination thresholds (3-log units and 13-log units

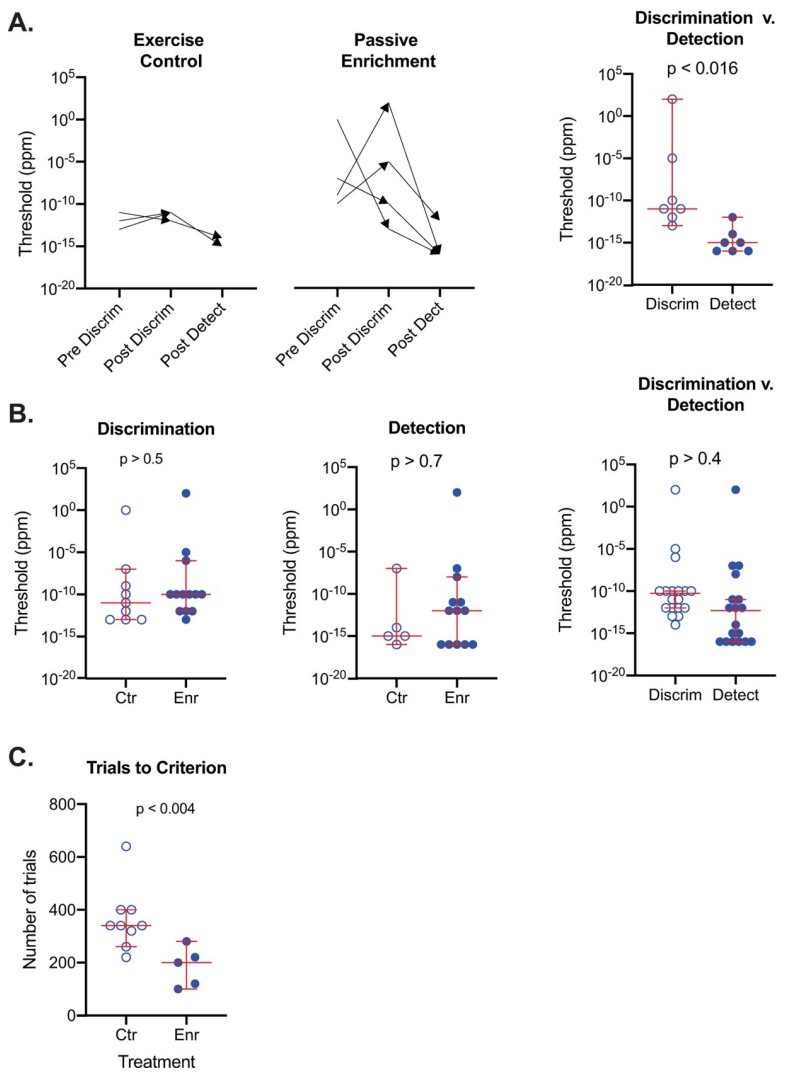

**Fig 3. Limonene threshold measurements of discrimination and detection for the pre/post and between-subjects designs. A.** Limonene discrimination (discrim) and detection (detect) thresholds for mice experiencing exercise control (Ctr) or passive odor enrichment (Enr). Pre and post values are depicted as the tails and heads of arrows to illustrate change in thresholds after enrichment. Since there was no obvious effect of odor enrichment, controls and experimental data were pooled to test whether discrimination thresholds were different from the detection thresholds. Median detection values were significantly lower than median discrimination values (Wilcoxon matched-pairs; W = -28; p < 0.016). **B.** Mice in pre/post and between-subjects designs were pooled to increase statistical power. Results are consistent with the conclusion that limonene enrichment did not have an effect on median discrimination thresholds (Mann Whitney; U = 47; p > 0.5) nor median detection thresholds (Mann Whitney; U = 32; p > 0.7). Since there was no effect of limonene enrichment on median thresholds, the two groups were pooled for analysis. The median limonene discrimination thresholds were not significantly different from median detection thresholds (Wilcoxon; W = -36; n = 18 pairs; p > 0.4). **C.** The median number of trials to reach criterion in the initial test of **R**-limonene stock vs. **S**-limonene stock were significantly different (Mann Whitney; U = 2.5; p < 0.004). For all figures, the long horizontal line = the median, short horizontal lines = 95% CIs.

respectively). As was the case for the exercise enrichment subjects, all four subjects in the passive enrichment group had lower thresholds for the detection task (ranging from 3-log units to 7-log units) than for the mixture discrimination task. The difference between subject performance in the mixture discrimination task and the detection task in the pooled exercise and passive enrichment groups was significant (Fig 3A; Wilcoxon matched-pairs; W = -28; p < 0.016).

## Limonene between-subjects design

The pre/post design was implemented at the beginning of these studies based on the assumptions that (1) there would be large between-subject variability in thresholds but consistent test-retest reliability within a subject and (2) that minimal odor exposure would be necessary to measure thresholds in the mixture discrimination test due to its presumed difficulty. Both of these assumptions turned out to be wrong. Thus, for all the other tests we changed to a between-subject designs with two groups: mice passively enriched or not enriched with odorant. This change in experimental design allowed us to do more replication and to minimize exposure to odorants during testing.

In order to increase the sample size of the limonene data set, a between-subjects group was added that consisted of five mice enriched passively by adding limonene to their food (see Methods and Materials) and two control mice that were not enriched with limonene. In addition, we used the post-data from the four subjects (M1-4) in the active enrichment group whose pretest data had been excluded. These subjects differed from passive enrichment mice in that they had extensive exposure to limonene in both the pretest and active enrichment treatment. That the active enrichment mice were (1) attending to the odors and (2) gaining additional experience with the operant task are borne out by their high average % correct responses over the 15 days of enrichment (M1 = 91.9%; M2 = 81.2%; M3 = 87.4%; M4 = 68.1%; binomial, $P(X \geq x)$ 0.04 for 60% correct responses in 100 daily trials).

Fig 4 shows the criterion responding graphs of these three groups, both for the mixture discrimination task and the detection tasks. First, it is apparent, as was the case for the data in Fig 2, that subjects tend to maintain very high rates of % correct responding right up to the stimulus concentration for which responding drops to chance levels. Second, some mice in each group achieved surprisingly low thresholds (down to $1 \times 10^{-14}$ ppm for mixture discrimination). And, finally, there is no evidence of induction. For example, post active-enrichment mice which had extensive exposure to limonene while performing daily operant tasks in the olfactometer, had thresholds similar to the control group that were not enriched.

To statistically evaluate these qualitative impression, all of the subjects tested prior to limonene enrichment, including the between-subject design controls, were pooled into a control group (n = 9) and all of the subjects tested after limonene enrichment, either active (n = 4) or passive (n = 9), were pooled into an enrichment group. Fig 3B (left) shows the group median and each subject's threshold so the variability of the measurements can be fully appreciated. The medians of the control and enrichment groups were not statistically different (Mann Whitney; U = 47; p > 0.5).

To test the effects of **R** and **S** limonene enrichment on **R**-limonene detection, pretest data for the exercise group (n = 3) were pooled with the between-subjects design control group (n = 2) to make an overall control group (n = 5). All subjects enriched with limonene actively (n = 4) or passively (n = 9) were pooled to form an overall enrichment group (n = 13). Fig 3B (center) displays the individual threshold values and the medians for the control group and the enrichment group. Note that the medians were not significantly different (Mann Whitney; U = 32; p > 0.7).

Since there was no evidence that limonene enrichment influenced either limonene mixture discrimination or **R**-limonene detection, all the discrimination thresholds were pooled and all of the detection thresholds were pooled to test whether there was a difference in the difficulty of these two tasks. Despite the significant results for the pre/post groups (see above), median discrimination threshold was not statistically different from median detection threshold (Wilcoxon; W = -36; n = 18 pairs; p > 0.4) despite the fact that the eight lowest thresholds were all in the detection group.

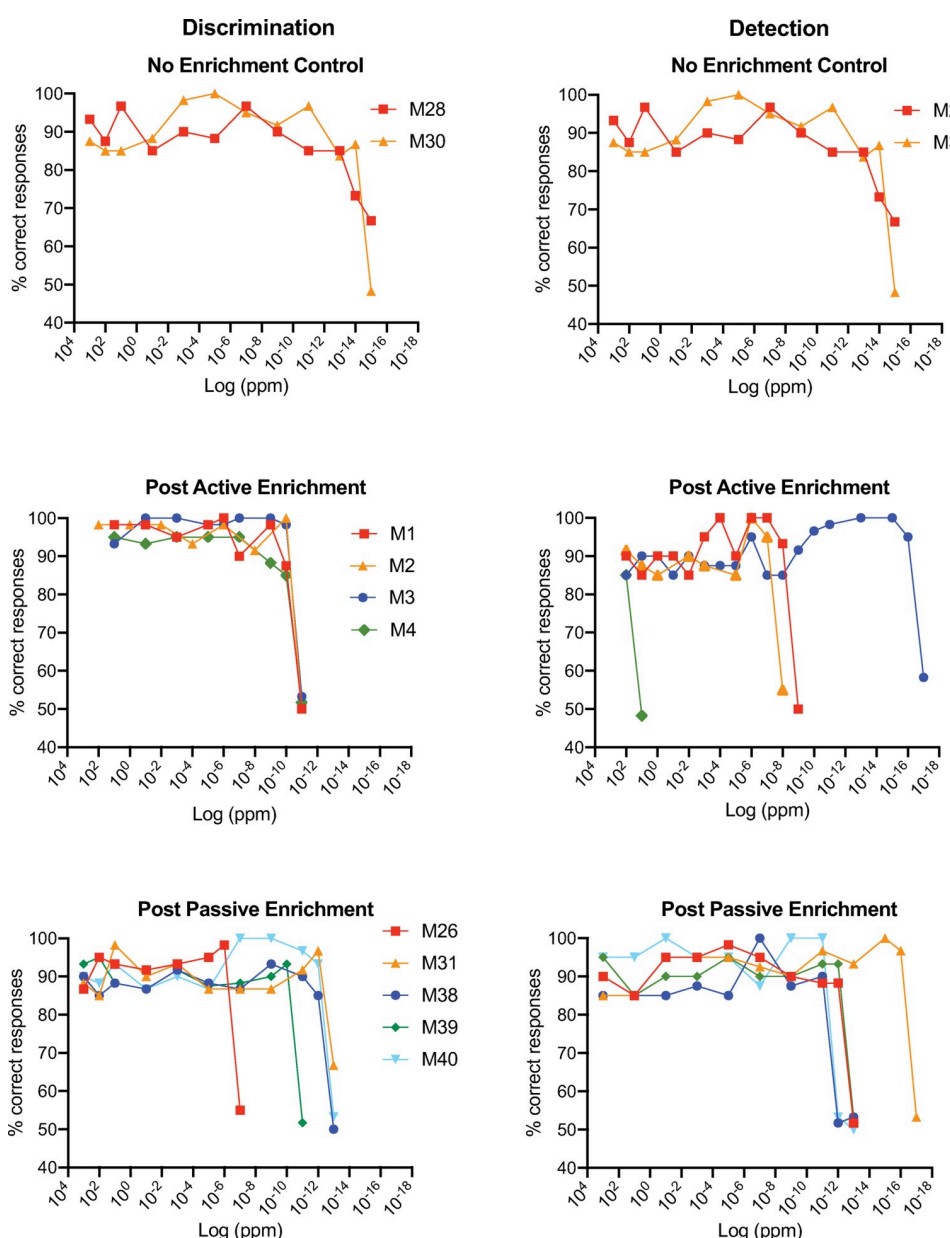

**Fig 4. Performance of mice discriminating or detecting various dilutions of limonene.** Discrimination (left column) was as described in Fig 1. and the stated concentrations are dilutions of **R**-stock in **S**-stock. Detection (right column) took the form **R** vs. mineral oil. Results are shown for mice that experienced no odor enrichment (Top), post-active enrichment (middle) and post-only passive enrichment (bottom). Note pre-active enrichment data were discarded (see Results). As above, each symbol represents the % correct responses for an individual mouse in the last two or three blocks of trials (see details of threshold criteria in Methods).

Finally, we reasoned that even if there was no effect of limonene enrichment on mixture discrimination or detection thresholds there still may be an effect on initial learning. We tested this hypothesis by tallying the number of trials needed to reach criterion (see Methods & Materials) for the initial discrimination of **S**-limonene stock-solution from **R**-limonene stock-solution. Unexpectedly, the median trial to criterion was significantly less for the enrichment group than the control group (Fig 3C; enrichment group = 200 trials; control group = 340

trials; Mann Whitney; U = 2.5; p < 0.004). Whatever the meaning of this result, we do not deem it an instance of induction since there was no difference in threshold. However, it does prove that there was some effect of enrichment.

## Carvone between-subjects design

The 18 subjects tested with carvone enantiomers, half in the control group and the other half in a passive enrichment group, were part of a between-subject design (see Methods). Shown in Fig 5 is criterion responding until threshold was reached for each subject with discrimination results (left) and detection results (right). As was the case for the limonene experiments, subject thresholds were highly variable spanning a 10-log-unit range for both discrimination and detection. And similar to the limonene experiments, some mice in each group had very low thresholds down to 1 x $10^{-8}$ ppm. However, these plots do not show any consistent differences between either the control and enrichment groups or between discrimination and detection in terms of threshold. To further evaluate these qualitative impressions, individual thresholds for each subject and the group median are plotted in Fig 6A. For discrimination (left plot), the median threshold for the enrichment group was substantially less than that for the control group. However, there was considerable overlap in the sample values and this difference was not significant (Mann Whitney; U = 20; p > 0.07). With respect to detection (Fig 6A, middle),

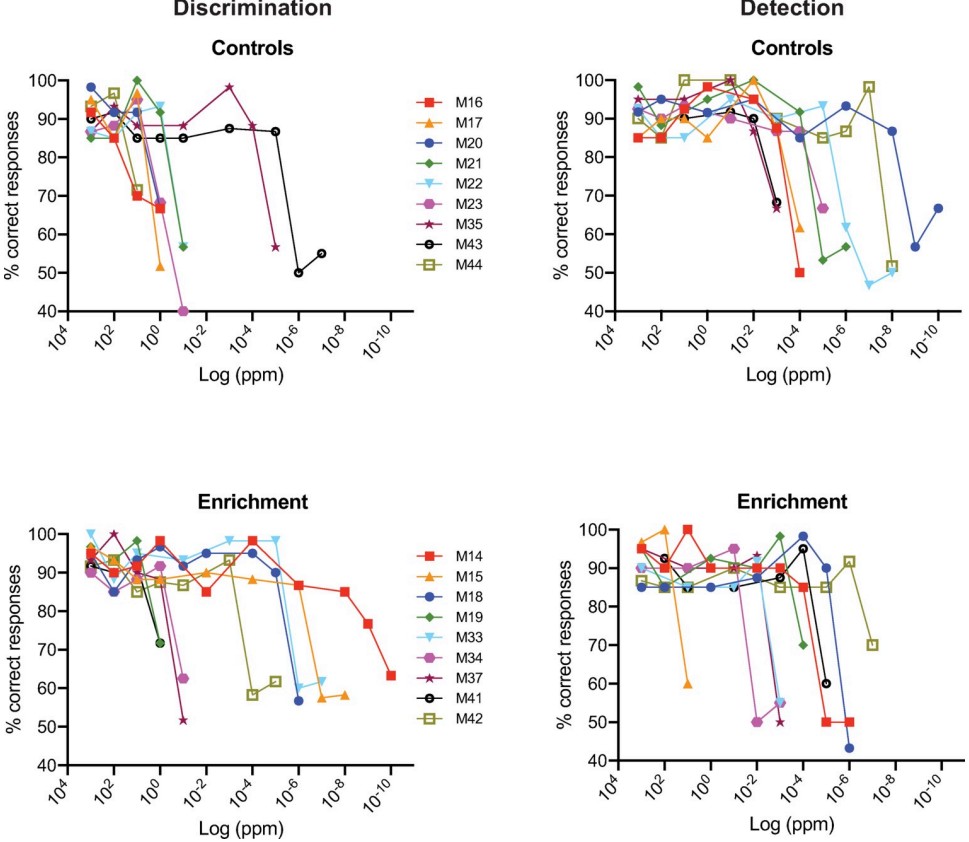

**Fig 5. Performance of mice discriminating (left columns) or detecting (right column) various dilutions of carvone.** Discrimination and detection were as described above. Results are shown for control mice that experienced no odor enrichment (Top), and mice that were passively enrichment (bottom). As above, each symbol represents % correct responses for an individual mouse in the last two or three blocks of trials (see details of threshold criteria in Methods and Materials).

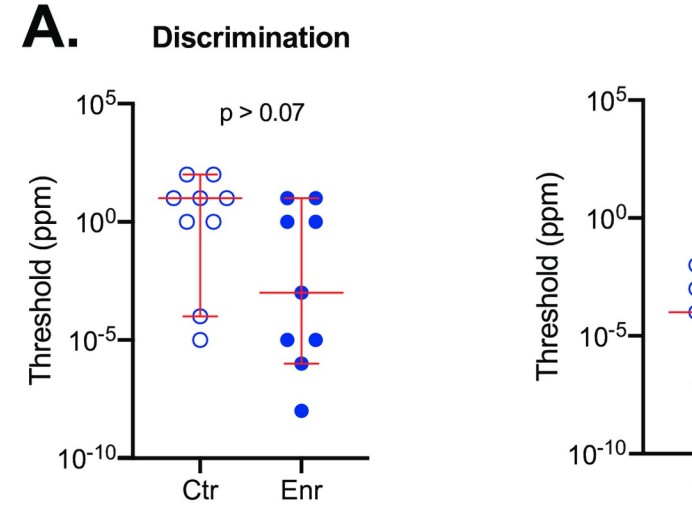
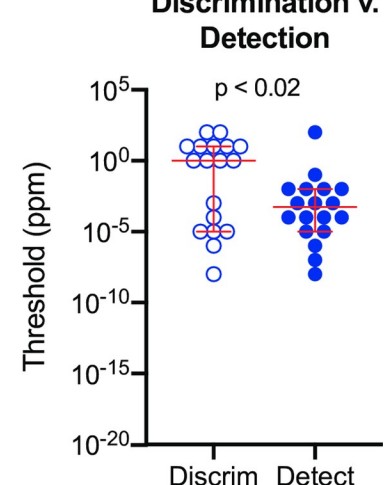

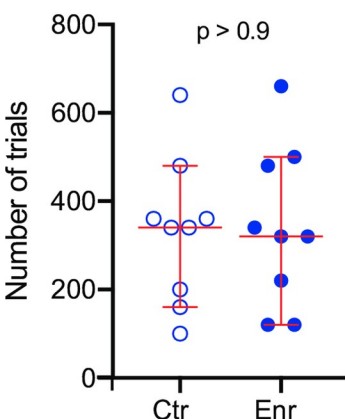

**Fig 6. Carvone discrimination and detection thresholds for mice experiencing no enrichment (Ctr) or passive enrichment (Enr). A.** Median discrimination (discrim) thresholds for controls were not significantly different from carvone enriched mice (left; Mann Whitney; U = 20; p > 0.07). Neither were the controls and enrichment detection (detect) medians significantly different (middle; Mann Whitney; U = 28.5; p > 0.3). Detection values were significantly lower than discrimination values as shown by comparing the pooled data from control and experimental mice (right; Wilcoxon; W = -95; n = 18 pairs; p < 0.02) **B.** The median number of trials to reach criterion in the initial test of **R**-carvone stock vs. **S**-carvone stock were not significantly different (Mann Whitney; U = 39.5; p > 0.9). For all figures, the long horizontal line = the median, short horizontal lines = 95% CIs.

the median threshold for the enrichment group was actually higher than that of the control group though not significantly so (Mann Whitney; U = 28.5; p > 0.3). The control and enrichment values were pooled within the discrimination and detection test groups since they were not significantly different (Fig 6A, right). The median threshold for the pooled discrimination data was significantly higher than for pooled detection data (Wilcoxon; W = -95; n = 18 pairs; p < 0.02) suggesting that discriminating **S** vs. **R** + **S** carvone is more difficult than detecting **R**-carvone.

Finally, we determined that the number of trials needed to reach criterion for the initial discrimination of the **S**-carvone stock solution from **R**-carvone stock solution were not significantly different (Fig 6B; enrichment group = 320 trials; control group = 340 trials; Mann Whitney; U = 39.5; p > 0.9).

## Vapor-phase concentrations

Owing to the difficulty of measuring vapor-phase stimulus concentrations at the point of subject contact in the olfactometer, most investigators report liquid-phase concentrations of the odor source [14, 16]. This was the approach adopted here given that our main goal was to compare psychophysical parameters with and without prior odor enrichment, not to definitively measure olfactory thresholds. However, in the process of carrying out these experiments it became apparent that we were obtaining thresholds among the lowest ever reported. Table 1 contains the median liquid-phase concentration thresholds for limonene and carvone discrimination and detection, their 95% confidence interval, and vapor-phase equivalents. Estimates of the vapor-phase concentration at the odor port were calculated using Henry's law parameter values measured for limonene by Cometto-Muniz and his colleagues [16]. We also took into consideration the 40-fold air dilution during the mixing of head space vapor in the stimulus bottles of the Knosys system with the carrier air that is channeled to the stimulus port. For example, the $5.5 \times 10^{-11}$ ppm median discrimination threshold for the limonene enantiomers is based on the liquid-phase concentration in the stimulus bottle of the **R**-enantiomer diluted in the stock solution of the **S**-enantiomer. This threshold becomes $6.16 \times 10^{-15}$ ppm after calculating the vapor-phase equivalent using Henry's law and dividing by air dilution factor of 40.

Our threshold data invite a number of conclusions. First, naïve mice are exquisitely sensitive to limonene and carvone enantiomers. Second, mice are extremely proficient in detecting the **R**-enantiomer in a background of the **S**-enantiomer. Indeed, they are only marginally better at discriminating the **R**-enantiomer from clean air (detection) than the mixture discrimination, though this difference is not statistically significant for limonene. Third, mice are ~ 9 log-units better at discriminating mixtures of limonene than mixtures of carvone (**S** vs. **R** + **S**) and mice are ~8 log-units better at detecting limonene than carvone. Finally, the thresholds measured in this study are among the lowest ever reported and we are among the few to report mixture discrimination values for limonene and carvone.

## Discussion

### Olfactory induction

The primary goal of this study was to test the hypothesis that odor enrichment alters odor acuity (used here to mean odor detection and discrimination), a process which has been termed induction. In a now classic study, Wysocki and colleagues showed that human subjects who were initially anosmic for the steroid odorant androstenone could become osmic for the odor through daily exposure over several weeks [10]. Mice, too, were later shown to have behavioral threshold improvements up to 10-fold after two weeks of daily exposure to amyl acetate or androstenone that were specific to the exposure odor [17]. These were most intriguing results not least because they suggested a novel form of sensory plasticity by which an animal's olfactory range adjusts to a new odor environment—without attention or feedback—rather than remaining static throughout life! Though clearly different, induction would seem to be related to phenomena considered olfactory forms of perceptual learning [18–20].

The locus (or loci) of induction—olfactory epithelium, olfactory bulb, or cortex—has not been fully explicated, but electrophysiological and lesion studies, in mice, suggest that the phenomenon is at least partly peripheral [17, 21, 22]. The EOG is an ensemble recording of generator potentials from olfactory sensory neuron (OSN) in the olfactory epithelium [23]. Mice from inbred strains that have characteristically low behavioral sensitivity to androstenone or isovaleric acid developed large increases in their electroolfactogram (EOG) responses to these odors after two weeks of 16 hour-per-day exposure. Subsequent studies from the same

laboratory found that the induction phenomenon is quite general with demonstrations of induction in normosmic mice in response to various odors including conspecific urine [24]. Induction was found to last for more than 6 months, in some cases, following acute episodes of odor enrichment [25].

A number of laboratories have demonstrated induction-like phenomena that may or may not have the same underlying physiology or adaptive (in the evolutionary sense) significance as the process described by Wysocki and colleagues [10, 17, 21]. For example, rabbits whose mothers have been fed juniper berries during gestation show behavioral preferences and enhanced EOG responses postnatally to juniper odorants compared to the offspring of unexposed controls [26]. And, a host of different associative-learning-dependent odor response enhancements have been reported including ones involving epigenetic effects [18, 27–30]. In these instances, cellular and circuit level plasticity in olfactory epithelium, bulb, and cortex have been variously implicated. Whether these associative processes form a different category of phenomenon from induction has been questioned [18].

While not strictly analogous to the induced odor detection studies of Wysocki and colleagues, discussed above, we chose to investigate the behavioral aspects of discrimination induction with the enantiomers of limonene and carvone. Previous studies in rodents found that discrimination of the former but not the latter stereoisomers could be induced by passive exposure. In several studies, naïve rats or mice failed to dishabituate to one stereoisomer of limonene after habituating to the other, suggesting they were unable to discriminate these mirror-molecules [13, 20, 31, 32]. However, after ten days of passive odor exposure, subjects spontaneously (without reinforcement) came to discriminate the stereoisomers of limonene in habituation-dishabituation tests (ibid.). We also wanted to use enantiomeric pairs because it was assumed that mixture discrimination would be highly difficult and thus lead to rapid threshold determination with a minimum of odorant exposure during testing. Finally, the use of enantiomers assured that discriminations would not be based on physical differences in odorants, like vapor pressure, mucus solubility, etc.

## No induction of limonene or carvone acuity

Contrary to the literature reviewed above and our assumptions, we found no evidence of induction using either passive (non-operant) or active (operant) enrichment regimes for either limonene or carvone. On the contrary, we found that naïve, CD-1 strain mice are amazingly proficient at discriminating a target member of an enantiomeric pair diluted in a background of the other member of the pair from the background stereoisomer alone (of the form **R** + **S** vs. **S** where **R** is the target). In our initial discrimination tests, acuity was so great, particularly for limonene, that it took numerous concentration steps to reach threshold using the method of descending limits. This situation prompted us to dispense with our pre/post experimental design in favor of an independent-samples design for nearly half of the mice tested with limonene and all of the mice tested with carvone in order to limit odor exposure during the act of threshold testing. Importantly, neither the results from subjects experiencing the pre/post testing nor the independent-sample design suggested any effect of odor enrichment with limonene or carvone on thresholds, though we cannot rule out the possibility that odor exposure during testing was, itself, inducing. In recognition of this possibility we considered implementing maximum likelihood methods of threshold determination because they require fewer trials to reach criterion. However, this option was rejected on the basis of a previous study which found that maximum likelihood estimates resulted in four-fold higher thresholds compared to stair-case procedures [33].

Our null findings resonate with seminal studies that employed longer (> 2 months) and stronger odor exposure regimes than those used here (albeit with different odors) which failed to observe any effect of odor enrichment on thresholds for the enrichment odorant despite the use of sophisticated olfactometric techniques in highly competent hands [34–36]. Unfortunately, there have been few other attempts at assessing the effects of odor enrichment on olfactory acuity in animals since these early studies. However, the literature abounds with reports of anatomical and physiological effects of passive odor enrichment that may fit within the induction framework. Focusing just on the olfactory epithelium, 20 days of continuous or "pulsed" odorant exposure caused mice to display *decreased* EOG responses, compared to controls, that were specific to the enrichment odor [37]. Consistent with this finding, feeding mouse dams heptaldehyde-laden food throughout gestation and the preweaning period also leads to a *reduction* in the magnitude of their pups' EOG responses to that odor as well as a reduction in transcripts of a heptaldehyde-sensitive olfactory receptor [38]. Still other investigators have found no effect of postnatal odor enrichment on EOG responses despite observing a decrease in the density of OSN subtypes expressing the olfactory receptors for which the enriched odor was the ligand [39]. This latter finding has been replicated with different ligand/receptor pairs [40, 41].

More confusing still are a number of seemingly contradictory studies demonstrating increased longevity of OSN subtypes that express an olfactory receptor for which the enriched odor is the ligand [30, 42–45]. Such OSN population selection should, over time, lead to larger amplitude EOG responses to the enrichment odor. Thus, it is surprising that no effect or a decrementing effect on the EOG have been the most common results of enrichment. Finally, it has been repeatedly shown that odor deprivation, the opposite of enrichment, leads to enhancements in olfactory responsivity, transductory pathways and acuity [reviewed by 46]. Taken together, our analysis of the conflicting and perplexing literature, limited to the effects of enrichment on the olfactory epithelium, hardly makes our inability to find induction surprising. The extensive literature on olfactory bulb and cortical plasticity following odor enrichment with or without associative learning will not be reviewed here in the interest of brevity. However, this literature also defies clear understanding with enrichment sometimes causing enhanced and sometimes diminished responses centrally that were either specific or non-specific for the odor used in enrichment, depending on the study [41, 47–49].

An impediment to consolidating the disparate results of the odor enrichment literature are the myriad odors, concentrations, schedules and modes of enrichment (vapor, food, drinking water) that have been used in different studies [11, 13, 17, 21, 24, 25, 31]. Perhaps our results would have been different had we enriched subjects for a longer period, though our enrichment durations were longer than those used by most other investigators that have observed induction. Alternatively, a shorter period or a different schedule of exposure might have produced an effect. Only four of our mice (active group) experienced truly intermittent odor enrichment through vapor-only exposure. These were the mice enriched with odor during the performance of the operant discrimination task in the olfactometers. The other mice in this study were odor enriched by mixing odor into their food. While previous studies have shown the utility of food or water for odor enrichment, the schedule and magnitude of odor exposure produced by these routes is unknown [48–50]. Our goal in using food as the vehicle for enrichment was to enforce regular but discontinuous odor delivery since the latter schedule might lead to chronic receptor adaptation [51]. We reasoned that subjects would be exposed to the enrichment odor, primarily, by a retro-nasal route during eating. And, the odors would gain salience by food association (classical conditioning). However, the fact that the food was available *ad lib*, and thus constantly available to evolve odor, could have led to chronic receptor adaptation. If this were the case, we might have expected a decrease in olfactory thresholds for the enriched odor.

Despite these limitations, the current results combined with our analysis of the contradictory literature lead us to the conclusion that induction may have less ecological relevance than originally envisioned [10, 17, 21]. Odor enrichment has not led to consistent improvements in olfactory acuity, increases in OSN sensitivity, or proliferation of cognate olfactory receptor proteins. In fact, one can find support in the literature for almost any prediction of odor enrichment's effects including: no effect, enhanced responses to the enrichment odor, or diminished responses to the enriched odor at whatever neural level of interest. And the effects of enrichment might be either specific to the enriched odor or generalized based on the reviewed studies. Importantly, the original suggestion that OSN clonal selection might underpin induction has not been vindicated by recent RNAseq analysis showing meager and equivocal effects of manipulating odor environment on olfactory receptor transcripts [50]. The logical benefit (i.e. adaptive significance) of spontaneously increasing sensitivity to odors in the environment that lack relevance or salience has been questioned on the grounds that nervous systems seem designed for exactly the opposite function: to filter out such unchanging stimuli [46]. In this respect, we submit that the analogy between induction and perceptual learning are inapt given that the latter phenomena are only rarely impactful when reinforcement and attention are lacking [7, 8, 11, 13, 20]. Rather, we suggest that at least some cases of induction may be laboratory artifacts produced by prolonged and unnatural exposures to high concentrations of odor that produce receptor adaptation, the release from which triggers a compensatory rebound response, though we observed no such effect here [46]. Undoubtedly, in nature, receptor adaptation is beneficial for stimulus normalization; it is the use of high concentrations—typically with a single purified odorant—and the long duration of most enrichment regimes that we deem artifactual [52]. The observation that induction, in the cases where it is seen, can be rapidly and spontaneously reversed adds support for this idea [39]. By contrast, odors which gain salience through associative learning mechanisms, garnering attention through rewarding and aversive contingencies, undoubtedly alter acuity through plastic processes at multiple levels of the nervous system [18, 19, 27–30]. It is a form of this latter process —supervised perceptual learning—not induction, which dominated the results of this study as will be discussed next.

## Olfactory acuity measurements

Automated, operant-based olfactometers (O-O), like the instruments used in this study, are generally considered to be "unparalleled" for the measurement of olfactory acuity in rodents [53]. However, these computer-assisted instruments require considerable experience, regular maintenance—including fastidious cleaning—and a great deal of time to obtain valid psychometric data [14]. These characteristics and the expense of the instruments, likely explain why they are not used more frequently in the assessments of olfactory acuity; and why, when they are used, surprisingly small sample sizes (~four to six subjects) are often on offer [e.g. 12, 54–57]. By contrast, the limonene and carvone thresholds reported here form one of the largest data sets of its kind with a total of 36 trained mice, 18 for each of the two test odors. Testing included both mixture discrimination and detection measures. In addition, for limonene, 11 of the subjects had thresholds determined both before and after stimulus exposure. Despite failing to find any evidence of induction, our methods allowed us to measure among of the lowest odor thresholds ever reported for any species, a fact which warrants further explication.

Fig 7 contrasts the threshold measurements from this study with other published detection thresholds for carvone, limonene and a few other selected odorants [54–57]. This compilation is by no means comprehensive, focusing largely on studies of mice that have employed the Knosys system or, in two other cases, similar O-Os [54, 58]. Added for further context are: (1)

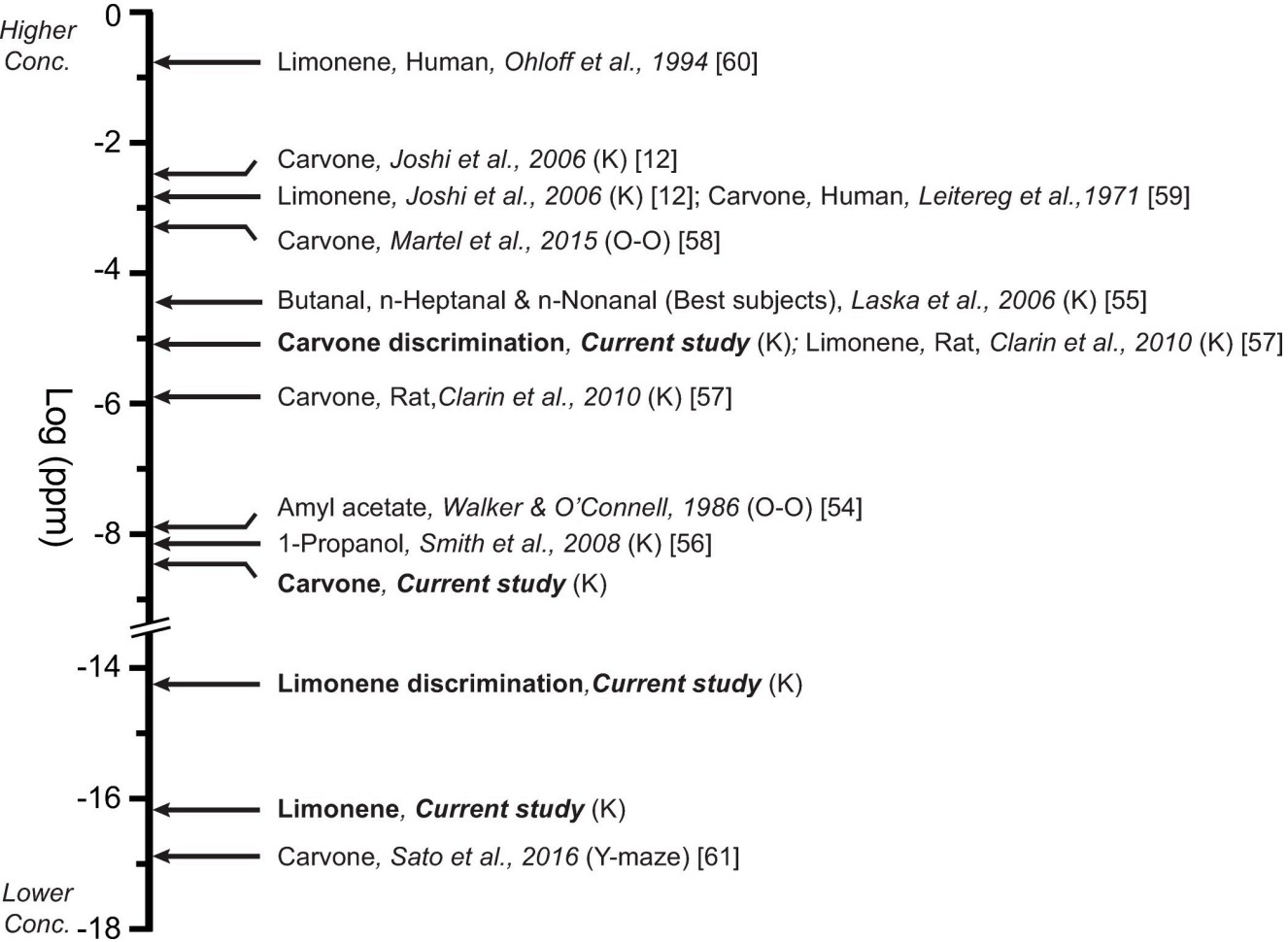

**Fig 7. The thresholds from the current study are compared to published thresholds for a select number of odors.** If not otherwise stated, all values are for detection tasks and represent means or medians. Values are in vapor-phase concentrations of log ppm. Where vapor phase values were not reported they were estimated using Henry's law for limonene or Raoult's Law for carvone [14, 16]. Selected values focus predominantly on limonene and carvone thresholds in mice and rats obtained using the Knosys system (K) or a similar operant olfactometer (O-O). For comparison, human thresholds are included and a few other exemplary reports. The human values were obtain using "squeeze bottles." Note that a break has been placed in the axis to save space.

the results of a rat study using the Knosys system; (2) a particularly noteworthy report using a Y-maze [59]; and (3) examples of human threshold reports for limonene and carvone are included [60, 61]. Perhaps the most troubling fact to be noted is the exceedingly wide span of results, even for identical odors within one species, the mouse. For example, detection thresholds for carvone (stereoisomer differences are ignored in this discussion) in mice range over more than 13 log-units [cf. 12, 59]. Granted, there are some differences between these extreme reports including strain of mouse, method of testing (O-O vs. Y-maze), and duration of training (days versus months). However, even if we limit comparisons to Knosys olfactometer studies in CD-1-strain mice, we find more than a six log-unit threshold disparity for carvone detection and a more than 13 log-unit disparity between limonene detection thresholds [cf. 12 and current study].

Though the irreproducibility of olfactory thresholds within and between studies has been frequently noted [62–64], the comparisons shown in Fig 7 are nevertheless startling. Indeed, they raise concerns about the validity of animal olfactory threshold, if not as a theoretic construct, at least as an empirical target. Of course, threshold measurements are only as reliable as

the appropriateness of the methods, the precision of the instruments, and expertise of the investigators measuring them. For example, as noted previously, naïve mice and rats have repeatedly been shown to be unable to "discriminate" the stereoisomers of limonene using habituation/dishabituation tests though they can readily be shown to do so in O-Os even within the first few trials (current study discrimination threshold = 6.16 x $10^{-15}$ ppm) [12, 57]. These discrepancies suggest that the habituation/dishabituation method, while possibly useful in some contexts, is clearly not a valid way to determine olfactory acuity [65].

In the case of the current study, we used the Knosys system, probably the most widely used O-Os for performing olfactory psychophysical testing (Fig 7); we were extensively trained in the instrument's use by the inventor, Dr. Slotnick; and we replicated our results to an almost unprecedented extent. Despite these advantages, we found marked variability in threshold measurements—more than 16-log units in some cases (see Figs 3 and 6)—a microcosm of the variability confronted when comparing olfactory thresholds among laboratories (Fig 7).

We believe several factors explain the irreproducibility of mouse O-O threshold studies: First, there is no standard for the operational definition of threshold ranging from quite liberal criteria (binomial test significance or ~ 60% correct depending on number of trials; [12] to quite stringent criteria such as 85% correct responses averaged over some number of trials (this study). As should be obvious, the more forgiving the % correct criterion the lower the threshold measurements that will be obtained. Second, there is no consistency in the other threshold criterion: the number of sessions and blocks of trials in which the subject is allowed to fail before meeting the % correct criterion. In some studies, subjects have been allowed as few as one session of 100 trials [12, 55], in other studies three sessions of 100 trials or more have been allowed [cf. 57 and current study]. Logically, the more attempts the subject is allowed, the lower the threshold down to some unknown limit. We assert that the reason we were able to measure such low thresholds was, in part, because we permitted the subjects the most blocks in which to meet criterion (340 trials separated in 17 blocks across three sessions) of any study of mouse thresholds that we have found in the literature (Fig 7). To quantify this assertion, we simulated what the thresholds would have been for a subsample of the mice in this study had we changed criteria. The thresholds for the group of mice discriminating limonene enantiomers that were part of the between-subjects experimental design would have been 13.1 (SEM ± 1.4; n = 7) log-units higher, on average, had we allowed them only one block of 100 trials and set the % correct criterion at 60% (cumulative binomial probability $P(X \geq x)$ 0.04) as some investigators have done [12, 55]. Thresholds from the active enrichment group pretests (see Results) were discarded precisely because we discovered, after the fact, how impactful this criterion was while testing other mice. A dearth of practice is likely the reason that maximum likelihood methods result in thresholds that are several log-unit higher than when the descending method of limits is used [33].

Interestingly, correct responses typically fell to chance levels each time odor concentration was lowered as if the mice were treating each concentration step like a qualitatively different stimulus. This need for the subject to relearn the discrimination when faced with identical chemicals at a different intensity is reminiscent of the task specificity observed in most forms of perceptual learning [6–8]. In a related vein, practice with an "easy" operant task over three weeks, the regimen of the active enrichment group, had no effect on final thresholds for these mice.

The third factor which we believe is important in explaining the variability in individual threshold measurements in our data and that of other investigators is maintaining subject motivation. Here the published protocols are fairly standard: water deprivation pegged to ~15% drop in body weight from the *ad lib* state which, for mice, means a 1 to 1.5 ml ration of water per day [33, 66]. However, despite assiduous efforts to maintain body weight at 85% of the *ad lib* state, subject motivation from day to day was quite variable in our study. Overly motivated

(thirsty) mice would sometimes default to licking on every trial presumably using the strategy that they would get rewarded on 50% of trials without having to make difficult discriminations (see Fig 1B; 1x $10^{-7}$ concentration). Unmotivated mice, some with body weights well below the critical mark, would take long pauses between trials sometimes lasting several minutes resulting in testing sessions that would last for hours. If a fraction of subjects were consistently overly motivated and others were consistently unmotivated, it would be a simple matter to exclude them from the study, but subjects often changed their motivational status over a period of days or even from session to session. For these reasons, we believe that medians, the midpoint of the lower-quartile range or even lowest thresholds (for examples) would be more meaningful measures of a species' olfactory acuity than average thresholds across an entire sample of subjects, which will often be compromised by the inclusion of unmotivated subjects.

These methodological issues aside, most of the mice in this study achieved thresholds for carvone and limonene discrimination and detection in the "super-sensitivity" range. Interestingly, Sato and colleagues [59] recently measured similarly low thresholds, in mice, for carvone and wine lactone enantiomers. Their exceptional results may be attributable to the use of a Y-maze, an instrument which affords subjects the simultaneous choice of S+ and S- stimuli and consequently may tap into a more natural behavior than a go/no go task. Critically, our measurements and those of Sato and colleagues [59] were carefully controlled for "cheating": the possibility that subjects may base their discriminations on non-olfactory cues (Fig 1C; see Control Procedures).

The low thresholds we measured serve as a rejoinder to the revisionist view that human olfaction is comparable to rodents despite the fact that the latter group has approximately four times the number of functional receptors as the former [67, 68]. In the case of limonene detection, mice in this study were more than 15 log-units more sensitive than humans based on published thresholds [61]. Many would agree that across species acuity comparisons are a fraught proposition, especially when one of those species is human [12]. But what about acuity testing in experimental settings, say with knockout mice or specific human-disease models? It could be argued that as long as the same methodology is used in the experimental and control groups, it is immaterial that absolute differences in olfactory threshold measurements among studies are irreproducible. In our view, the problem lies in understanding what the results of such studies mean. Clearly, the thresholds reported here and those by Sato and colleagues [59] suggest that mice—still the primary tool of transgenic research—have much greater odor acuity when the capabilities are fully realized than previously appreciated. This fact raises the possibility that many previous studies showing differences in odor acuity between experimental and control groups were actually tapping into motivational or perceptual learning differences rather than peripheral or other lower level processes. Reports of negative findings are particularly ambiguous since our study suggests mice in some previous studies may simply have been undertrained [56, 58]. Obviously, the neural processes that subserve motivation and perceptual learning are likely to be quite different from those for peripheral functionality such as OSN expression and physiological responses [69].

Lastly, though testing models of olfactory coding was not a goal of this project, the fact that enantiomer discrimination was no more difficult than detection in the case of limonene and only slightly more difficult for carvone deserves comment. Limonene's receptors have not been studied in detail to our knowledge. But carvone interacts with chiral-specialist and chiral-generalist olfactory receptors, with the latter outnumbering the former by two-to-one in the mouse [59, 70]. Despite this preponderance of chiral-generalist receptors, our results suggest that the encoding of the enantiomers of carvones, which have different odor qualities for humans, must rely exclusively on the specialist; otherwise detection would have a lower threshold than discrimination [59, 70].

## Conclusions

The ability of mice to discriminate or detect the stereoisomers of two mirror-molecules, limonene and carvone, were unchanged by either passive or active enrichment with these odors despite the use of sensitive olfactometric measurements and large sample sizes. This result contrasts with previous claims that such an induction process is a general olfactory phenomenon, particularly for enantiomer discrimination. Nevertheless, dramatic improvements in discrimination and detection emerged when mice were allowed more than the typical allotment of trials to reach criterion on a particular concentration during threshold testing. We suggest that this provision of extra practice, compared to that allowed in most previous studies, is the reason we were able to measure among the lowest olfactory thresholds reported for any species. Notably, most concentrations in the descending method of limits used to find thresholds were initially treated by subjects as qualitatively different stimuli, as responding typically fell to chance levels before rebounding. This task specificity is reminiscent of that reported for perceptual learning processes in vision and audition. Also of interest, subjects displayed statistically equivalent (limonene) or only modestly higher (carvone) acuity in a detection task compared to a discrimination task suggesting receptors that are chiral-specialists dominate acuity performance near threshold. The profound effect of supervised (with feedback) perceptual learning on subject performance in this study undermines any notion that olfactory thresholds are merely a readout of receptor or low-level processes, an implicit assumption of many studies using gene targeted mice. However, the super-acuity for limonene and carvone displayed by the mice in this study compared with the dramatically lower acuity of humans for these odors, reported by others, challenges the proposition that human cognition and other human endowments compensate for rodents' four-fold larger receptor repertoire. Finally, our results suggest that supervised-perceptual learning—rather than passive induction—is the form of plasticity what allows a species' olfactory system to achieve its maximum potential.

## Acknowledgments

The authors thank Burt Slotnick for scientific advice and technical assistance throughout the course of this study.

## Author Contributions

**Conceptualization:** David M. Coppola.

**Data curation:** Alyson Blount, David M. Coppola.

**Formal analysis:** David M. Coppola.

**Funding acquisition:** David M. Coppola.

**Investigation:** Alyson Blount, David M. Coppola.

**Methodology:** Alyson Blount, David M. Coppola.

**Project administration:** David M. Coppola.

**Resources:** David M. Coppola.

**Supervision:** David M. Coppola.

**Visualization:** David M. Coppola.

**Writing – original draft:** David M. Coppola.

**Writing – review & editing:** Alyson Blount, David M. Coppola.

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
