## [Decision Letter · Decision Letter 0]

5 Jun 2020

PONE-D-20-12663

The effect of odor enrichment on olfactory acuity:  Olfactometric testing in mice using two mirror-molecular pairs.

PLOS ONE

Dear Dr. Coppola,

Thank you for submitting your manuscript to PLOS ONE. Apologies for the delays - I had to just step in as an academic editor as the original editor became unable to handle the manuscript.

After careful consideration, we feel that it has merit but does not fully meet PLOS ONE’s publication criteria as it currently stands. Therefore, we invite you to submit a revised version of the manuscript that addresses the points raised during the review process.

It will be key to address the methodological concerns raised by the two reviewers, notably ideally a control or at least detailed discussion of potential residual odor problems (as raised by reviewer 2) and ideally additional data on the reliability of serial dilutions (raised by reviewer 1) or a thorough discussion. Additionally, more details on the statistics is needed. Please see also the other very valuable comments and recommendations of the two expert reviewers. 

We look forward to receiving your revised manuscript.

Kind regards,

Andreas T. Schaefer

Academic Editor

PLOS ONE

Journal Requirements:

Reviewers' comments:

Reviewer's Responses to Questions

**Comments to the Author**

1. Is the manuscript technically sound, and do the data support the conclusions?

Reviewer #1: Partly

Reviewer #2: Partly

2. Has the statistical analysis been performed appropriately and rigorously? 

Reviewer #1: No

Reviewer #2: Yes

3. Have the authors made all data underlying the findings in their manuscript fully available?

Reviewer #1: Yes

Reviewer #2: Yes

4. Is the manuscript presented in an intelligible fashion and written in standard English?

Reviewer #1: Yes

Reviewer #2: Yes

5. Review Comments to the Author

Reviewer #1: Review of PONE-D-20-12663 “The effect of odor enrichment on olfactory acuity: Olfactometric testing in mice using two mirror-molecular pairs.”

This manuscript sets out to test the ability of unsupervised learning through a process known as induction to improve detection and discrimination thresholds of enantiomeric pairs of odors. Throughout the study mice report the presence of odors using a standard go/no-go behavioral task. As part of this task the authors use an interesting experimental approach for their discrimination assay. Here they ask mice to discriminate a pure odor enantiomer and another that has been diluted with the enantiomer of opposing chirality. Although the experiments in this manuscript fail to find evidence for induction, the authors report detection and discrimination thresholds that exceed many prior reports.

I have two major concerns that I will also expand on below. First, the justification for pooling data in many instances is not well described. In the least, I hope that the authors will provide statistical analysis accompanying data prior to pooling. Second, a technical concern, the authors claim to report the among lowest threshold measurements ever recorded for any species. While I agree that the data do indeed look promising, several important technical controls including vapor phase odorant measurements are needed to support such a claim. Below I provide a point-by-point critique:

1) The choice of enantiomers seems justified but is somewhat curious given that enantiomers of both carvone and limonene are perceptually dissimilar, therefore it is not clear whether there is anything special about enantiomeric pairs, with regard to odor discrimination. In the discussion the authors point out that enantiomers interact with chiral-specialist and -generalist receptors; however, this is also true for any odors that activate overlapping patterns of OSNs - of which there are many. I recommend a general deemphasis regarding the special case of enantiomers. However, one key advantage that might be mentioned is that the vapor pressure should be nearly the same between enantiomers.

Related to the above point in Line 516: “We also wanted to use enantiomeric pairs because it was assumed that mixture discrimination would be highly difficult” – This may be a false assumption especially given that it is well documented that carvone enantiomers are perceptually dissimilar and activate distinct (yet overlapping) populations of ORNs.

2) My primary concern regarding statistics is the justification for pooling data that used different experimental conditions. My main issue is that the treatment between the two groups is fundamentally different and therefore it may not be appropriate to group data together. I request that the authors provide a more detailed description as to their justification. This should extend beyond a lack of effect in either group alone. A formal power analysis might be considered here.

Related to the above point in Line 311: While I do not have a strong objection to the authors pooling the data from the two experiments and allowing readers to draw their own conclusions, I do think that it is necessary to present statistics for each the datasets prior to pooling. In essence these are two fundamentally different experiments (passive odor exposure vs exercise), therefore, at face, it is not intuitive why these data should be pooled.

3) Figure 1: please provide an x-axis in these plots.

4) Figure 2: These plots might be more effective if the data for each mouse is plotted on the same axes for each of the three conditions. Also related to these data, were the mice only tested once at each odor concentration, or is the data at each point an average of several sessions? If so, please provide error bars.

5) Figure 3: It is very hard to follow where the n values come from throughout this figure. Please revise the text to add clarity and justification when data was pooled.

Figure 3: I am also generally confused about the message of the last panel in part A. If detection occurs at significantly lower thresholds than discrimination after enrichment, are these values indistinguishable prior to enrichment? Some other measures might be informative for the interpretation of this data, for example, plotting the change in threshold for discrimination/detection with exercise/enrichment.

6) Major technical concern: The dilutions used are very small and may be impacted by the viscosity of mineral oil when transferring between serial dilutions. Did the authors use a device such as a photoionization detector to measure vapor phase odor concentrations? Without such measurements to control for dilution-to-dilution variability, it is hard to be certain that the authors are truly delivering the odor concentrations that they calculate. Furthermore, it appears that the claim of the low detection/discrimination thresholds are based on data from a number of mice, but it unclear whether these were single blocks of trials in each animal or if the low thresholds could be reliably measured across several days/sessions of behavior. Can the authors please explain if these data are from single blocks of trials or are averages of several sessions.

7) Line 331: Can it really said that the null-hypothesis was rejected for the pre/post experiments given that statistical power is very low with n = 3 and 4? This is especially true for the passive enrichment group given the large variance across replicates.

8) Perhaps the authors can explain why the detection/discrimination curves all seem to fall to chance (50% accuracy) at a specific dilution of the odorant. A reasonable assumption is that there should be some performance degradation that appears as the animals begin to approach detection threshold. Often times data such as these are plotted as a psychometric function that can be fit with a hill equation. In this case, it appears the data would be better approximated by a step function. Can the authors please provide some insight as to why performance drops off all at once?

Reviewer #2: Blount and Coppola have provided a rigorous, well-reasoned, and scholarly study, with two major results: 1) "odor enrichment" or "induction", though it causes various changes in the olfactory epithelium in bulb, and has been shown by others to affect thresholds, does not induce a detectable change in thresholds when measured thoroughly in well-trained mice. 2) well-trained mice can have remarkably low thresholds, not just for odor detection but for discrimination of enantiomers. I find result 1 convincing, and in my view this result alone provides sufficient reason to accept this study for publication. However, I have misgivings about result 2.

Water restricted mice are devious experimental subjects (not that I blame them). The authors have convincingly demonstrated that their mice were not using non-odor cues to guide their choices. However, I'm not sure the authors have eliminated the possibility that residual odor cues could be contributing to the behavioral performance they observe. In my lab, we have found in several assays that mice can perform in olfactory tasks even when presented with nominally blank vials. After all too much troubleshooting, we found that the problem was residual odor -- this performance would drop to chance after a thorough cleaning of the olfactometer. The authors performed separate "0-0" control sessions, but the problem is that residual odor can accumulate sporadically over sessions. Some days the residuum may have accumulated to a detectable level, other days not. This may explain some of the across-mouse and across-day variability in performance. In my view, the best way to properly control for this possibility is to interleave "0-0" trials into every session (see eg, Findley et al, biorXiv 2020). Dewan et al (2018) had to go to great lengths to avoid this effect (see Methods (incidentally, this study also found extremely low thresholds, and may be worth including in figure 7)) -- no-odor trials were presented from multiple blank vials, some of which treated as S+ others as S-. Importantly, mice were evaluated for "cheating" in every behavioral session.

My paranoia may be misplaced in the case of this study. I am not familiar with the Knosys olfactometer design, and it may have features that eliminate this possibility. It would be helpful to include a schematic of the design so that this can be evaluated. Otherwise, I encourage the authors to discuss whether or not an extraneous olfactory cue that varies from day to day can be excluded as a contributor to the remarkably low thresholds they observe for limonene.

6. PLOS authors have the option to publish the peer review history of their article (what does this mean?). If published, this will include your full peer review and any attached files.

Reviewer #1: Yes: Joseph D. Zak

Reviewer #2: Yes: Matt Smear

---

## [Author Response · Author response to Decision Letter 0]

3 Jul 2020

Please see attached letter entitled Response to Reviewers

---

## [Decision Letter · Decision Letter 1]

15 Jul 2020

The effect of odor enrichment on olfactory acuity:  Olfactometric testing in mice using two mirror-molecular pairs.

PONE-D-20-12663R1

Dear Dr. Coppola,

We’re pleased to inform you that your manuscript has been judged scientifically suitable for publication and will be formally accepted for publication once it meets all outstanding technical requirements. Please have a look at reviewer 1's suggestions that you might want to include when providing the final manuscript. 

Kind regards,

Andreas T. Schaefer

Academic Editor

PLOS ONE

Additional Editor Comments (optional):

Reviewers' comments:

Reviewer's Responses to Questions

**Comments to the Author**

1. If the authors have adequately addressed your comments raised in a previous round of review and you feel that this manuscript is now acceptable for publication, you may indicate that here to bypass the “Comments to the Author” section, enter your conflict of interest statement in the “Confidential to Editor” section, and submit your "Accept" recommendation.

Reviewer #1: All comments have been addressed

Reviewer #2: All comments have been addressed

2. Is the manuscript technically sound, and do the data support the conclusions?

Reviewer #1: Yes

Reviewer #2: Yes

3. Has the statistical analysis been performed appropriately and rigorously? 

Reviewer #1: Yes

Reviewer #2: Yes

4. Have the authors made all data underlying the findings in their manuscript fully available?

Reviewer #1: Yes

Reviewer #2: Yes

5. Is the manuscript presented in an intelligible fashion and written in standard English?

Reviewer #1: Yes

Reviewer #2: Yes

6. Review Comments to the Author

Reviewer #1: Review of PONE-D-20-12663_R1, “The effect of odor enrichment on olfactory acuity: Olfactometric testing in mice using two mirror-molecular pairs”.

This revised submission makes substantial changes that improve the manuscript in clarity and content. The authors have provided a thorough response to each of my prior critiques and following their clarifications I support the publication of this work. Major changes include new statistical information related to pooling datapoints that satisfies my prior concerns. There are also confidence intervals added to figures where comparisons have been made between groups.

My other major concern regarding odor dilutions has been at least partially addressed and I am willing to accept the author confidence in their serial dilutions, now bolstered by simulations. However, given that the technical limitations of vapor phase odor concentration measurements and the extremely low detection thresholds reported in this study, I encourage that the authors report the total number of mice that achieved the 10-16 ppm threshold for limonene, as well as the number of trial blocks that each mouse achieved this extremely low threshold.

Minor: Figure 3 is missing a label for part B.

Reviewer #2: From the authors' response, I see that they had already anticipated and addressed my concern. The added description of their controls will hopefully allow readers to appreciate the care that is necessary in performing psychophysics on the sensitive nose of the devious mouse. I hope the rigor shown here will be emulated by the rest of the field.

7. PLOS authors have the option to publish the peer review history of their article (what does this mean?). If published, this will include your full peer review and any attached files.

Reviewer #1: **Yes: **Joseph D. Zak

Reviewer #2: **Yes: **Matthew C Smear

---

## [Editor Report · Acceptance letter]

20 Jul 2020

PONE-D-20-12663R1 

The effect of odor enrichment on olfactory acuity:  Olfactometric testing in mice using two mirror-molecular pairs. 

Dear Dr. Coppola:

I'm pleased to inform you that your manuscript has been deemed suitable for publication in PLOS ONE. Congratulations! Your manuscript is now with our production department. 

Kind regards, 

on behalf of

Dr. Andreas T. Schaefer 

Academic Editor

PLOS ONE